# Exploring Potential Complement Modulation Strategies for Ischemia–Reperfusion Injury in Kidney Transplantation

**DOI:** 10.3390/antiox14010066

**Published:** 2025-01-08

**Authors:** Dario Troise, Costanza Allegra, Luciana Antonia Cirolla, Silvia Mercuri, Barbara Infante, Giuseppe Castellano, Giovanni Stallone

**Affiliations:** 1Nephrology, Dialysis and Transplantation Unit, Advanced Research Center on Kidney Aging (A.R.K.A.), Department of Medical and Surgical Sciences, University of Foggia, 71122 Foggia, Italy; 2Division of Renal Medicine and Baxter Novum, Department of Clinical Science, Intervention and Technology, Karolinska Institutet, 141 52 Stockholm, Sweden; 3Unit of Nephrology, Dialysis and Transplantation, Fondazione IRCCS Ca’ Granda Ospedale Maggiore Policlinico di Milano, 20122 Milan, Italy

**Keywords:** kidney transplantation, complement system, ischemia–reperfusion injury, Eculizumab, iptacopan, complement-targeting drugs

## Abstract

The complement system plays a crucial role in regulating the inflammatory responses in kidney transplantation, potentially contributing to early decline in kidney function. Ischemia–reperfusion injury (IRI) is among the factors affecting graft outcomes and a primary contributor to delayed graft function. Complement activation, particularly the alternative pathway, participates in the pathogenesis of IRI, involving all kidney compartments. In particular, tubular epithelial cells often acquire a dysfunctional phenotype that can exacerbate complement activation and kidney damage. Currently, complement-modulating drugs are under investigation for the treatment of kidney diseases. Many of these drugs have shown potential therapeutic benefits, but no effective clinical treatments for renal IRI have been identified yet. In this review, we will explore drugs that target complement factors, complement receptors, and regulatory proteins, aiming to highlight their potential value in improving the management of renal IRI.

## 1. Introduction

The kidney transplantation process involves a series of events that can impact the survival of the graft, including brain or cardiac death in deceased donors, organ procurement, preservation, and reperfusion. Moreover, post-transplant immune responses and non-immune factors, including drug toxicity, hypertension, and recurrent diseases, may have a pivotal role affecting the outcomes of the kidney graft. Increasing evidence indicates that the complement system is a key regulator of both inflammatory and non-inflammatory responses throughout each stage of transplantation, potentially leading to the early decline of kidney function [1].

Due to the complex relationship between oxygen consumption, oxidative stress (OxS), renal blood flow, and glomerular filtration rate, the kidneys are considered among the organs most susceptible to hypoxic injury. Ischemia–reperfusion injury (IRI) is considered one of the main factors involved in the pathogenesis of acute kidney injury because it leads to cellular damage, which can impact outcomes, affect prognosis, and potentially limit the success of the kidney transplantation (KT) [2]. IRI occurs when a temporary loss of blood supply to the transplanted organ (ischemia) is followed by the restoration of blood flow and reoxygenation. IRI is a complex process in which hypoxic tissues are exposed to ischemic damage, marked by cellular dysfunction and death, while the subsequent reperfusion paradoxically intensifies tissue damage by amplifying the inflammatory response. The complement system participates in the pathogenesis of IRI, and its activation is considered one of the main components of renal IRI. Consequently, therapeutic strategies to prevent and treat renal IRI by targeting the complement system are being actively explored [3]. Currently, kidney damage resulting from IRI is a leading cause of delayed graft function (DGF) following KT, impacting short- and long-term graft survival [4].

This narrative review aims to emphasize the role of the complement system in renal IRI, with a focus on therapeutic strategies that target its modulation.

## 2. The Complement System

The complement system is considered an evolutionarily conserved component of the innate immune system characterized by an intricate network of molecules that play a crucial role in maintaining tissue homeostasis and host defense [5]. Over the years, a growing number of experimental evidence has demonstrated that the complement system has a crucial role because it carries out fundamental physiological functions, including pathogen and cell debris removal, the clearance of immune complexes, and the recruitment of components of the adaptive immune system [6]. In addition, crosstalk between the complement system and other systems and pathways has been found, including the adaptive immunity system [7], the coagulation system [8], the inflammasome [9], and toll-like receptors (TLRs) [10].

The number of identified components in the complement system is steadily increasing, with more than sixty components known to date. In addition to the nine central components of the cascade (C1 to C9), the complement system includes soluble proteins, membrane-bound receptors, and regulatory proteins [11,12]. Notably, while most complement proteins are produced in the liver and circulate largely in an inactive form, the kidney, brain, lung and intestines are among the few organs capable of locally synthesizing complement proteins [13]. In 1991, Brooimans et al. showed that human proximal epithelial tubular cells can synthesize C3 when stimulated by IL-2 [14]. Recently, the kidneys have been identified as the primary sources of complement proteins, with mesangial, endothelial, epithelial cells and podocytes being capable of synthesizing the circulating complement proteins C3, C4, and factor B and complement factor H (CFH) [15].

Complement activation is triggered by the recognition of pathogen-associated molecular patterns (PAMPs) or damage-associated molecular patterns (DAMPs), achieved via different types of pattern recognition molecules (PRMs). Based on the signaling pathway initiated, the complement cascade can be activated through one of three main pathways: the classical pathway (CP), alternative pathway (AP), or lectin pathway (LP) [16,17].

Each pathway has distinct mechanisms for the recognition of activation signals. In the CP, the recognition of immune complexes by C1q is necessary for initiation. Then, this signal leads to the subsequent activation of C1r and C1s, subsequently cleaving C4 and C2 into small fragments (C4a and C2a) and large fragments (C4b and C2b). The large fragments then combine on the cell surface to form the C3 convertase (C4bC2b). In contrast, the LP is triggered by the binding of specific pattern recognition receptors (PRRs) to PAMPs or DAMPs, leading to the activation of MBL-associated serine proteases (MASPs), the cleavage of complement components C4 and C2, and the subsequent formation of C3 convertase.

Unlike the other two pathways, the AP utilizes different C3 convertases. Priming of the AP relies on a process known as “tickover”, where the circulating C3 molecule undergoes spontaneous low-rate hydrolysis in the fluid phase. Following the binding of C3 to factor B cleaved by factor D into Ba and Bb fragments, the generation of the C3 convertase of the AP (C3Bb) occurs [18]. Interestingly, the C3b generated by the classical and lectin pathways, C3 convertase, can be utilized to form the C3 convertase of the alternative pathway. Moreover, proteases not typically classified as complement components, such as neutrophil elastase and thrombin, can also cleave C3, resulting in an amplification of the alternative pathway [19].

Both the convertases cleave C3 into C3a and C3b, resulting in the formation of C5 convertase (C4b2b3b or C3bBb3b of the CP and AP, respectively), which then cleaves C5 into C5a and C5b. The latter contributes to the assembly of the membrane attack complex (MAC) alongside C6, C7, C8, and C9. Thus, all three pathways converge on the formation of the MAC and unfold sequentially. Target cells are directly lysed by the formation of the MAC, which creates 10 nm pores in the cell membrane, exerting direct cytolytic effects and leading to the release of a large number of cellular components, further activating PRMs and intensify inflammatory responses [18].

Additionally, other effector molecules, such as opsonins (C4b, C4d, C3b, iC3b, C3dg, and C3d) [20] and anaphylatoxins (C3a, C5a, and C4a), are generated as a result of complement activation. Opsonins are deposited on the cell surface and tag the cell for phagocytosis by binding to various complement receptors, a process referred to opsonization. In contrast, anaphylatoxins induce vasodilatation, histamine release, and the chemotaxis and activation of mostly macrophages and neutrophils by binding to their specific receptors, further amplifying the inflammatory response [21].

By interacting with their respective receptors—CR1, CR2, CR3, and CR4 for opsonins and C3aR, C5aR1, and C5aR2 for anaphilatoxins—these molecules can recruit other inflammatory cells to the infection site and modulate the activity of immune cells [18]. In addition, anaphylatoxin has been shown to influence endothelial cell proliferation and migration as well as cancer development [22] (Figure 1).

Moreover, under physiological condition, there are surface-bound and soluble proteins that control complement activation. These complement regulator proteins include C1esterase inhibitor (C1INH), factor H (FH), factor I (FI), decay-accelerating factor (DAF or CD55), and membrane cofactor protein (MPC or CD46); they act mainly through decay acceleration and cofactor activity. In addition, surface-bound regulators are relatively nonspecific, regulating all three complement activation pathways and inactivating both C3 and C4. In contrast, soluble regulators are more specific, targeting only the alternative, classical, or lectin pathway, and acting exclusively on either C3 or C4 [23].

An additional pathway of complement activation involving the coagulation system has also been identified. Some components of the coagulation cascade, including thrombin, activated factors XI, X, and IX, and plasmin, have been shown to mediate the proteolytic breakdown of complement components and directly activate C3 and C5, independently from the C3 convertase, leading to the introduction of the concept of a fourth complement activation pathway, known as the “extrinsic complement pathway” [24,25,26]. On the contrary, complement components have been shown to induce tissue factor expression and the activation of platelets or endothelial cells, which leads to the secretion of the von Willebrand factor, establishing a positive feedback loop that may serve as a powerful amplification mechanism in IRI [27,28,29]. Moreover, signaling crosstalk between the complement system and immunological pathways involved in the coordination of immune and inflammatory responses, such as the toll-like receptor (TLR) pathway, has been associated with increased inflammation and host tissue injury, when dysregulated. For example, common microbial molecules, including lipopolysaccharide (LPS) and bacterial CpG DNA, can act as both TLR ligands and complement activators. In particular, LPS may induce the release of factor B of the alternative pathway in macrophages. Furthermore, epithelial and innate immune cells can be locally stimulated by TLR activators to produce key components for alternative complement activation, thereby amplifying TLR-mediated responses [30]. Among the TLRs, TLR-4 has gained attention due to its pivotal role in the pathogenesis of renal IRI. Increases in both TLR-4 expression and ligands were observed in tubular epithelial cells (TECs) and vascular endothelial cells after kidney transplantation, leading to the activation of downstream signaling pathways involved in the promotion of TEC injury and contributing to sustained inflammation and fibrosis [31]. Recently, C3 gene knockdown has been shown to downregulate the TLR-4 signaling pathway in a mouse model of renal IRI, mitigating the damage and leading to the stabilization of the podocyte cytoskeleton [32].

To maintain homeostasis, the human complement system carefully balances activation and inhibition, ensuring that the intensity of the reaction is appropriately regulated. Moreover, the complement system is activated in several kidney diseases, including glomerulonephritis, vasculitis, and tubulointerstitial diseases. However, significant differences exist between the microenvironment in the glomerulus and that of the tubulointerstitial compartments. Proper complement regulation is crucial at the glomerular level to protect the kidney from the constant exposure to these proteins. In fact, all surface-bound complement regulators, including CD46, CD55, and CFH, are expressed on resident glomerular cells, as these cells are continuously exposed to high concentrations of complement proteins and immune complexes [33]. Contrarily, tubular epithelial cells are involved mainly in the activation of the complement system in response to injury. Thurman et al. demonstrated that the loss of polarity of complement receptor 1–related protein Y in the tubular epithelium after ischemic injury resulted in the activation of the alternative pathway in this compartment [34].

It is important to note that complement inhibition can be associated with serious infectious complications. Particularly, the deficiency of complement activation or terminal components has been associated with Neisseria meningitidis infection. However, the widespread use of vaccines targeting all the major invasive serotypes with vaccination prior to the initiation of complement inhibitors can significantly reduce the infection risk. Depending on the complement target, vaccination can provide different levels of protection, revealing the need for the close monitoring of vaccine efficacy over time [35]. Theoretically, the inhibition of the terminal components of the complement cascade, such as C5, preserve C3-associated effector functions, including cell opsonization and immune cell activation, leading to a lower risk of infectious complications compared to upstream C3 inhibition. Moreover, the prophylactic use of antibiotics and other antimicrobial agents active against common opportunistic pathogens, including encapsulated bacteria and fungi, can help to reduce the risk of infection [36].

Additionally, a temporary inhibition of the complement system could be a workable option, because it has been demonstrated that long-term complement inhibition may increase the risk of adverse events, while short-term inhibition is unlikely to pose significant problems [37]. Furthermore, considering the effects of complement inhibition therapies, in order to assess their efficacy as well as adverse events, it is advisable. This can be achieved by measuring circulating levels of specific complement components or activation products, or through the assessment of the functional capacity of particular pathways, in order to adjust drug dosages based on the patient’s immune status, complement activity levels, and infection susceptibility, helping to identify and mitigate potential adverse events [38].

## 3. Oxidative Stress and Complement System

Oxygen deprivation followed by hyperoxygenation during IRI leads to the rapid generation of reactive oxygen and nitrogen species (ROS and RNS), causing rapid tissue injury and contributing to the overall inflammatory burden. During the ischemic phase of IRI, the absence of oxygen triggers anaerobic metabolic cell pathways, compromising the synthesis of ATP, which is considered the aerobic energy source of the cells. Then, during the reperfusion, the sudden influx of oxygen exceeds the cell’s capacity to manage it. Both of these phases lead to the generation of OxS.

The mitochondria are considered the primary source of ROS due to the dysfunctional activity of mitochondrial complexes I and III and the subsequent ROS production that overwhelms the antioxidant defenses of the cells. Additionally, OxS can also arise from other cellular sources, including xanthine oxidase, NADPH oxidase, and endothelial nitric oxide synthase; these sources activate key pathways involved in oxidative stress responses, including the hypoxia-inducible factor (HIF) and nuclear factor erythroid2-related factor2 (NRF2) pathways [39,40]. Therefore, the excessive production of ROS impairs mitochondrial function by damaging the respiratory chain, altering mitochondrial membrane permeability, and inducing mutations in mitochondrial DNA. These effects lead to a breakdown in energy production and a progressive loss of mitochondrial efficiency [41].

The existing literature indicate that OxS can activate the complement system through all three complement pathways [42,43,44]. Adler et al. were among the firsts to explore the mechanisms by which MAC deposited in the glomerular mesangium could be a trigger. They stimulated rat glomerular mesangial cells with MAC derived from the purified human complement component C5b6 and measured the production of ROS, showing that MAC could induce the formation of both superoxide ion and hydrogen peroxide in mesangial cells. Interestingly, cells exposed to decayed MAC showed only mild effect or no increase in ROS production, suggesting a correlation between the deposition of MAC and OxS [45]. Moreover, Choi et al. investigated the relationship between ROS production in human umbilical vein endothelial cells (HUVECs) exposed to OxS-induced damage and complement activation. Their findings revealed, once again, that the deposition of the MAC was associated with ROS production, and treatment with the antioxidant N-acetylcysteine significantly reduced complement activation [43]. Moreover, OxS has been shown to be correlated with complement activation in studies where HUVECs were incubated with plasma from patients with thrombotic microangiopathy, demonstrating an increase in ROS production, apoptosis, and elevated C3 levels [46].

Anaphylatoxins are involved in promoting cellular responses and host defense. They have been reported to bind to their specific receptors localized on both the plasma membrane and inside the cells and correlate with OxS levels. In retinal pigment epithelial cells exposed to OxS, C3aR is internalized and can be transported to mitochondria by endosomal trafficking, where it elevates mitochondrial calcium levels and impairs respiratory function. Furthermore, the C5a/C5aR signaling axis can affect mitochondrial morphology by inducing fragmentation and an increase in the number of pyknotic nuclei. OxS also upregulates the expression of mitochondrial fusion-related proteins, mitofusin-1 and mitofusin-2. This suggests that both anaphylatoxins may play a role in sensitizing cells to ROS-induced damage [47,48]. These findings reveal the role of cellular oxidative strain in complement system-mediated injury.

Conversely, intracellular signals triggered by the complement system can stimulate ROS production and cause mitochondrial damage, resulting in the increased release of mitochondrial DAMPs. It has been shown that the intracellular activation of C5aR1 by C5a can promote the production of ROS in CD4^+^ T cells, facilitating the assembly of a functional inflammasome system [49]. Moreover, Tsai et al. observed that C5a promotes ROS production in murine kidney endothelial cells. Elevated levels of C5a/C5aR induced cell apoptosis through cytochrome C release and caspase 3/9 activation, a process dependent on ROS generation, suggesting that C5a inhibition could directly reduce OxS and improve mitochondrial function [50].

In human proximal tubular cells, C3a has been shown to induce NOX enzymes, which are involved in the NADPH-dependent reduction of oxygen to generate superoxide, and the upregulation of alpha smooth muscle actin expression, which is a specific marker for fibrotic myofibroblasts. In an animal model of IRI, silencing NOX4 RNA resulted in decreased alpha smooth muscle actin expression, the suppression of NADPH activity, and a reduction in overall oxidative burden [51].

Collectively, these findings highlight the pivotal role of the complement system in driving OxS through a bidirectional relationship. The activation of the complement system may lead to the production of ROS, while OxS can, in turn, activate the complement system (Figure 2).

In addition, several antioxidant agents have been proposed for the treatment of IRI, including compounds that target nuclear factor erythroid 2-related factor 2 (Nrf2), hydrogen sulfide, dexmedetomidine, edaravone, and other mitochondria-targeting antioxidants specifically designed to provide their delivery inside the mitochondria [52]. Some of them appear to be involved in the regulation of the complement system. Particularly, heme oxygenase-1, known as a key regulator of many pathophysiological processes, including apoptosis, cryoprotection, and inflammation, has recently been identified as a regulator of the complement control protein decay-accelerating factor. Natural and synthetic compounds, such as metalloporphyrins, which can modulate the expression of heme oxygenase-1, hold potential for the development of cost-effective and non-toxic strategies for the management of complement system-mediated diseases [53]. Furthermore, complement component 3 has been associated with proinflammatory effects in an animal model of IRI. Yang et al. showed that the treatment with an antioxidant compound, such as N-tert-butyl-α-phenylnitrone, reduced C3 expression, thereby reducing IRI-related damage [54]. In inflammatory and ischemia-associated diseases, complement components can cause the adhesion of leukocytes to the endothelial wall. This is considered one of the major mechanisms responsible for the production of oxygen-derived free radicals, cytotoxic oxidants, and inflammatory mediators. In this context, the oral administration of flavonoids has been demonstrated to reduce total serum complement levels and the number of adherent leukocytes, highlighting the protective effects of these compounds in mitigating proinflammatory conditions, such as IRI [55]. Therefore, strategies that modulate the complement system in conjunction with antioxidant and anti-inflammatory compounds may have a potential impact in the mitigation of complement system- and OxS-related damage during IRI.

## 4. The Complement System in Renal Ischemia–Reperfusion Injury

IRI is frequently an unavoidable consequence that can lead to increased morbidity and mortality in various other clinical scenarios, including myocardial infarction, stroke, vascular surgery, and trauma. The organs that may be affected by IRI include the lungs, heart, gut, brain, and, notably, the kidneys. It is important to note that, in addition to transplantation, several other factors can trigger IRI in the kidneys, such as vascular and cardiac surgeries, sepsis, trauma, and cardiocirculatory arrest followed by resuscitation [56,57] (Figure 3). Moreover, while IRI can occur in kidney grafts from living donors, it is generally more severe in organs from deceased donors. Pre-existing medical conditions, hemodynamic insults, and extended warm and cold ischemia times account for complement system activation after the release of DAMPs and inflammatory responses. Furthermore, deceased donors have increased levels of C5a and C5b-9, correlated with an increased risk of rejection compared to living donors [58,59].

Metabolic changes in cells and the loss of cellular homeostasis are major players driving the pathways involved in IRI. During the transplantation process, the kidneys are exposed to phases of ischemia and reperfusions. The term “warm ischemia” refers to the time when an organ remains at body temperature after its blood supply has been interrupted but before it is cooled or reconnected to a blood supply, while “cold ischemia” is the period between cold perfusion and blood supply regeneration. Prolonged cold ischemia adversely affects the long-term survival and functional recovery of transplanted organs. Moreover, severe oxygen deprivation leads to a shift to anaerobic metabolism across all compartments of the renal parenchyma, resulting in ATP depletion, intracellular acidosis, increased production of reactive oxygen species (ROS), and the release of DAMPs. These changes play a crucial role in activating innate immunity, particularly the complement system, as well as adaptive immunity [60].

Significant alteration in renal proximal tubular epithelial cells has been found to be associated with IRI due to its ability to drive the downregulation of complement system regulator factors and an increase in the local production of complement components from renal TECs, which are recognized as major targets of complement system activation [61], resulting in functional and structural changes, including the loss of brush borders, alterations in the cytoskeleton, and a rapid loss of surface markers that promote the activation of cell death mechanisms. Moreover, TECs acquire a dysfunctional phenotype, characterized by cell swelling, the increased expression of various cell adhesion molecules, and increased permeability, key factors contributing to prolonged kidney dysfunction. In addition, ischemia leads to interstitial inflammation and micro-vasculopathy, further exacerbating the damage [41,62].

Even if the CP has been traditionally associated with the pathogenesis of complement-mediated IRI [63], studies in animal models of renal IRI suggest that the AP and LP play a predominant role in in triggering the complement cascade during renal IRI, while the role of CP is limited. Zhou et al. evaluated renal structural and functional injury after IRI in mice, showing that C3-, C5-, and C6-deficient mice exhibited protection from the injury, while C4-deficient mice did not [4]. In addition, Thurman et al. showed that factor B-deficient mice, an essential component of the AP, developed significantly less functional and morphological injury after renal IRI than the wild-type mice, which exhibited an increased deposition of complement C3 and neutrophil infiltration, suggesting that the complement system in the kidneys is mainly activated by AP, and therapeutic strategies aiming to inhibit this pathway may offer protection against IRI [64]. Supporting this, later studies by Renner et al. in 2011 showed that the inhibition or lack of factor H, a negative regulator of the AP, worsened kidney injury [65]. Regarding the role of the LP, Møller-Kristensen et al. investigated its contribution to complement system activation in a bilateral renal IRI animal model, showing that mannan-binding lectin double-knockout mice exhibited less evident acute tubular necrosis and lower levels of C3a than wild-type mice, indicating reduced C3 deposition and overall complement activation [66]. However, recently, complement fraction C5a and membrane attack complex (MAC, C5b-9) have been found to be elevated in brain-dead donors, suggesting that downstream complement products also play a crucial role in mediating renal damage. Moreover, C5-b9 has been linked to a higher risk of kidney rejection following transplantation [67,68].

Additionally, several complement regulators and anaphylatoxins receptors have been identified as crucial in the development of IRI. Genetic deficiencies in CD55 and CD59, which are membrane-bound complement regulators, have been shown to increase renal IRI. Moreover, the modulation of their activity appears to increase the inhibition of the C3 and C5 convertases, thereby reducing IRI damage, as demonstrated by Bongoni et al., who showed that mice expressing both CD55 and CD59 exhibited improved renal parameters after experiencing moderate IRI [3]. In addition, further studies demonstrated that anaphylatoxin receptor inhibition, such as that of C3aR and C5aR1, expressed in both infiltrating immune cells and kidney parenchymal cells, with their expression levels increasing in response to IRI, prevented kidney IRI by reducing chemokine and cytokine expression and decreasing inflammatory cell infiltration [69,70,71].

Given the critical role of the complement system in renal IRI, identifying biomarkers or clinical indicators to select patients who might gain the most from therapies that modulate the complement system is essential for optimizing therapy and minimizing side effects. In this context, elevated levels of anaphylatoxins (C3a and C5a) and serum C5b-9 were observed 24h after IRI was correlated to organ damage. Moreover, Budkowska et al. demonstrated a positive correlation between C3a levels and C-reactive protein, an inflammatory marker, immediately after transplantation, suggesting that this relationship could represent a prognostic signature of allograft survival or rejection in an IRI liver transplantation model [72]. Elevated expression levels of complement-related mRNA were observed in a renal IRI animal model, where locally produced C3 mRNA was upregulated within 2h of reperfusion in ischemic rat kidneys, without the significant involvement of circulating C3 [73].

Additionally, significant evidence suggests that endothelial involvement in renal IRI contributes to a reduction in renal blood flow, impairing the vascular compartment of the kidney. Endothelial dysfunction may lead to retrograde complement activation, creating a vicious cycle of amplified inflammatory responses that exacerbate tissue damage and lead to organ failure [74,75]. Recently, endothelial-derived complement factor D has been associated with increased levels of endothelial dysfunction markers, including intercellular adhesion molecule-1 (ICAM-1), vascular adhesion molecule-1 (VCAM1), von Willebrand factor (VWF), and endothelin-1, suggesting that the complement factor D can mediate endothelial dysfunction [76].

Monitoring these markers could guide therapeutic approaches in order to tailor complement-targeting therapies to individual patients and optimize clinical outcomes.

## 5. Potential Therapeutic Strategies

An increasing number of complement system-modulator drugs have recently gained approval or are in the final stages of clinical testing for the treatment of inflammation-based conditions. Given the close relationship between complement activation and the resulting inflammatory response in renal IRI, numerous research groups have developed various therapeutic strategies to inhibit or reduce complement activation in order to mitigate kidney damage. In detail, we can arbitrarily distinguish three primary targets for intervention: complement C5 and C3, complement receptors/regulatory proteins, and complement factors B and D. The rationale behind complement system-modulating therapies in renal IRI is that targeting complement cascade factors directly blocks specific factors to prevent their activation and subsequent damage, and targeting receptors involved in the pathway leads to the disruption of the signaling and decreased recruitment of inflammatory cells, while acting on regulatory proteins could prevent the overactivation of the complement system.

### 5.1. Targeting Complement C5 and C3

Although there is currently no effective clinical treatment for renal IRI [77], a growing body of evidence indicates that complement activation at the C3 convertase and downstream C5 convertase stages plays a central role in mediating renal IRI, highlighting that targeting complement cascade factors might be a promising therapeutic target to help prevent damage in high-risk recipients [73,78].

One of the most targeted complement factors is C5 because it serves as the endpoint for all three pathways. Inhibiting C5 may effectively provide a terminal blockade without disrupting the functionality of upstream proteins, thereby minimizing the impact on basic functions of the complement, including opsonization and chemotaxis.

The first anticomplement drug approved by the US Food and Drug Administration (FDA) and the European Medicines Agency in 2007 for the treatment of paroxysmal nocturnal hemoglobinuria (PNH) [79] and later for Atypical Hemolytic Uremic Syndrome (aHUS), Generalized Myasthenia Gravis [80], and Neuromyelitis Optica Spectrum disorders [81] was the long-acting humanized anti-C5 monoclonal antibody (mAb) Eculizumab (Soliris^®^, Alexion Pharmaceuticals, Boston, MA, USA). However, in the field of transplantation, it has garnered significant attention for treating antibody-mediated rejection and for prophylactic use in preventing aHUS recurrences in kidney-transplanted recipients [82]. Moreover, studies are currently evaluating the efficacy of Eculizumab for the treatment of C3 glomerulopathy in kidney-transplanted patients, with promising results [83]. The mechanism of action of Eculizumab is based on the prevention of the cleavage of C5 into C5a and C5b fragments, thereby inhibiting the activation of the terminal complement pathway, including the formation of complement effectors C5a and the MAC [84].

Currently, the efficacy of Eculizumab is being investigated for several other conditions, including renal IRI in transplanted patients. Researchers are exploring whether the inhibition of terminal complement system activation could help reduce tissue damage, improve graft survival, and prevent complications in transplant patients. In 2013, Kaabak et al. described a case of renal IRI after a kidney transplantation from a deceased donor who met the standard donor criteria, where the renal IRI was completely reversed after a single infusion of Eculizumab [85]. This laid the foundation for a randomized controlled trial involving 57 pediatric kidney transplant recipients, in which half of the patients received a single dose of Eculizumab before transplantation. The results showed that the patients who received Eculizumab exhibited better early graft function, reduced arteriolar hyalinosis, and chronic glomerulopathy in early and late biopsy samples, compared to the control group [86]. However, Schröppel et al. reported data from two randomized studies regarding the use of Eculizumab in reducing DGF rate after kidney transplantation. They did not find differences between the group treated with Eculizumab and the control groups in terms of DGF rate and graft function. These results were in line with the larger phase II/III PROTECT (Prevention of Delayed Graft Function Using Eculizumab Therapy) study, which did not show significant differences in DGF rates or graft survival between the treatment and control groups [87]. The lack of efficacy of Eculizumab may be related to the specific inhibition of the downstream part of the complement pathway, without blocking the activity of upstream molecules which could be key mediators of DGF despite C5 inhibition. Future clinical studies will be needed to explore the combination of Eculizumab with other agents that regulate upstream complement system activation in order to combine their mechanism of actions.

Many pharmaceutical companies are developing alternative anti-C5 therapies inspired by the success of Eculizumab in treating various complement system-involved diseases. However, additional research is needed to confirm their effectiveness in managing renal IRI. For example, Ravulizumab (Ultomiris™, Alexion Pharmaceuticals, Boston, MA, USA) is a mAb, designated by the substitution of four amino acids in the eculizumab backbone in order to extend its half-life [88]. It is under study in the ARTEMIS protocol for the management of major adverse kidney events caused by ischemic injury, including IRI, in patients undergoing cardiac surgery [89]. Crovalimab (Roche/Genentech, South San Francisco, CA, USA), another mAb that targets an epitope on the C5 β chain (whereas Eculizumab and Ravulizumab bind to the C5 α chain [90]), is currently under investigation in a randomized phase Ib study for the management of acute uncomplicated vaso-occlusive episodes in sickle cell disease, a condition characterized by microvascular dysfunction associated with IRI [91].

Currently, one of the most important challenges in managing atypical hemolytic syndrome, which shares the common denominator of AP activation with IRI, is determining the appropriate timing for discontinuation of the therapy. A scientific consensus conference suggested that, once kidney function has improved and stabilized, discontinuation should be considered in patients without pathogenic variants in complement genes, and that close monitoring with blood tests and weekly urinary dispticks is essential to detect early relapses and restart the treatment [92]. Since renal IRI can exacerbate complement system activation and potentially increase the risk of hemolytic uremic syndrome relapses in kidney transplant patients, the prevention of IRI should be considered essential [93].

Another compelling target in the management of renal IRI is complement factor C3, because it plays a crucial role in all three activation pathways and is responsible for proximal complement activation. Currently, efforts are focused on developing drugs designed to inhibit C3. Among them, Pegcetacoplan (Apellis, Waltham, MA, USA), a pegylated anti-C3 cyclic peptide, has already been approved by regulatory agencies for the treatment of PNH in patients who do not respond adequately to Eculizumab. Pegcetacoplan is a compostatin derivative, a group of peptides that specifically bind to C3, preventing access to C3 convertases and thereby inhibiting C3 cleavage across all three complement activation pathways [94]. Moreover, there is significant interest among researchers in the field of kidney transplantation regarding the use of Pegcetacopan in the treatment of post-transplant recurrent C3 glomerulopathy or immune complex membranoproliferative glomerulonephritis [95]. Due to its small size (43.5 kDa), it may effectively reach the corticomedullary junction, a hypoxia-sensitive region of the kidney, and therefore potentially reduce the activation of the complement system during IRI [6]. In the future, clinical trials could be designed to assess the effectiveness of combining C3 and C5 inhibition in IRI to evaluate whether this combination might enhance efficacy compared to targeting C3 or C5 alone. Recently, evidence has emerged to suggest that for certain diseases, such as paroxysmal nocturnal hemoglobinuria (PNH), the simultaneous blockade of both the proximal and terminal complement pathways may be useful. In a review article by Notaro and al., the authors suggested a dual-inhibition approach to prevent C3 binding to red blood cells, thereby reducing extravascular hemolysis and preventing massive breakthrough hemolysis in PNH patients [96].

Currently, the first monoclonal antibody targeting both C5 and the C3 convertase is KP104 (Kira Pharmaceuticals, Cambridge, MA, USA); it provides both terminal and proximal complement pathway inhibition [97] and is currently under investigation in a phase II study to evaluate its efficacy and safety in patients with IgA nephropathy and C3 glomerulopathy [98]. Moreover, the potential application of KP104 in the context of renal IRI should not be ignored. The dual inhibition of the both proximal and terminal complement pathways may result in an oversuppressed fontline defense for the immune system; therefore, close monitoring of complement activation markers, prophylactic vaccinations, regular infection screenings, and regular assessment of the patient’s immune status may help to maintain immune balance in this patient population.

### 5.2. Targeting Complement Receptors and Regulatory Proteins

The risk of recurrent infections can be increased by the use of compounds that broadly inhibit complement system activation. Therefore, a more targeted approach in order to better modulate complement system activity may offer benefits, potentially leading to similar positive outcomes for patients. In this context, the inhibition of the C5a-C5aR1 axis does not interfere with MAC formation, but it could be particularly advantageous due to the powerful chemotactic and pro-inflammatory properties of this anaphylatoxin [99].

In animal models of renal disease, blocking C5aR1 resulted in reduced small vessel inflammation and neutrophils activation in ANCA-associated glomerulonephritis [100]. Furthermore, recent studies have demonstrated that interactions between C5a and C5aR1 play a critical role in the development of renal IRI and contribute to kidney rejection. Zang et al. demonstrated that the use of recombinant C5a led to a reduction in progranulin levels, a grow factor with anti-inflammatory properties relevant in renal IRI. However, C5aR deficiency resulted in reduced NF-kB expression after IRI, which reversed the C5a-induced suppression of progranulin, supporting that the complement C5a/C5aR pathway exacerbates renal IRI by downregulating protective factors and promoting inflammation [101]. According to this, Peng et al. demonstrated that C5a caused the upregulation of inflammatory-related genes, such as interleukin-1α (IL1α), IL-6, and transforming growth factor-α in RECs. The inhibition of the C5a-C5aR1 axis significantly reduced renal injury and tubulointerstitial fibrosis, suggesting a pathogenic role for C5aR1 in the progression of fibrosis after renal IRI [102].

A powerful antagonist of C5aR1, Avacopan, has been approved by the FDA for the additional treatment of severe active cytoplasmic antibody antineutrophil (ANCA)-associated vasculitis (AAV) in combination with standard therapy in adults; it significantly improves patient kidney function by reducing proteinuria and increasing the estimated glomerular filtration rate [103]. Furthermore, C5aR expression from RECs has been reported in several kidney diseases, including diabetes, IgA nephropathy, sepsis, and IRI, suggesting its relevance in renal pathology [104].

An alternative strategy to prevent C5a signaling involves its inhibition within circulation, Vilobelimab (IFX-1), is a monoclonal IgG4 kappa antibody that selectively binds to C5a, blocking its biological activity. Authorized by the FDA for emergency use in treating COVID-19 in hospitalized adults, it is also under investigation for use in other conditions, including granulomatosis with polyangiitis, hidradenitis suppurativa, and pyoderma gangrenosum [105]. Interestingly, recent findings indicate that COVID-19 infection is related to a decreased oxygen levels and the development of IRI not only in the lungs, but also in heart, liver, and kidneys, suggesting that targeting C5a might be beneficial in attenuating renal IRI [106].

Human complement receptor 1 (CR1) is an integral membrane protein expressed on the surface of several types of cells, including podocytes, that inhibits complement activation by preventing the formation of the convertases that activate C3 and C5. Additionally, a soluble form of CR1 is also present in the circulation. Both membrane-bound and soluble CR1 help regulate complement system activity by promoting the decay of the C3 and C5 convertases and serving as a cofactor for the serine protease factor I, which degrades C3b and C4b [107]. Kassimatis et al. explored the use of Mirococept, a recombinant complement inhibitor derived from human CR1, in the EMPIRIKAL trial to evaluate its effects on reducing DGF in kidney transplantation. Initial findings showed that the ex vivo delivery of Mirococept is both feasible and safe; however, a dose calibration is necessary [108]. Based on this, the EMPIRIKAL-2 trial is now revising its human study protocol based on the optimal higher dose achieved in pigs [109].

Although the AP is considered the main pathway responsible for renal IRI, recombinant human C1 esterase inhibitor (C1INH), a powerful inhibitor of proteases of the CP and LP, has been studied in the context of renal IRI to prevent the amplification of the inflammatory responses. Castellano et al. demonstrated that the infusion of C1INH into a swine model of renal IRI was associated with a significant C4d and C5b-9 deposit reduction in the peritubular capillaries and glomerular compartments, as well as decreased infiltration of inflammatory cells [110]. Later, Danobeitia et al. investigated the use of C1INH in a mouse model of IRI, showing that the use of C1INH before transplantation significantly improved survival rates and provided protection against subsequent kidney fibrosis [111]. However, in humans, a phase I/II trial investigated the efficacy of C1INH in reducing the need for hemodialysis during the first week post deceased-donor kidney transplantation (definition of DGF [112]). While no differences in DGF were observed between the groups, those who received C1INH required significantly fewer dialysis sessions 2 to 4 weeks post transplant, suggesting that C1INH may be useful to prevent IRI and DGF in kidney allografts, but larger studies will be needed [113].

Complement factor H is a glycoprotein primarily produced by the liver which circulates in a soluble form, reaching high plasma concentrations with a complement system-inhibitory function. The interaction of factor H with certain ligands, including glycosaminoglycans, sialic acid, and C3b fragments, can anchor it to the surface of cells and membranes, thereby providing alternative pathway regulation locally. Notably, mutations in factor H can make specific cell surfaces vulnerable to alternative pathway activation, potentially disrupting normal complement regulation in those sites [114]. Goetz et al. demonstrated that in mice with a heterozygous deletion of the factor H gene there was the development of more severe kidney damage after IRI than in wild-type controls. However, complement activation was restricted to the tubulointerstitial compartment and did not occur in the glomeruli, suggesting that full factor H expression may be not necessary for controlling glomerular complement activation after IRI and that its modulation could be useful for the management of IRI-associated damage [115]. Currently, clinical trials are testing the adeno-associated virus-mediated delivery of a fusion protein that combines a complement receptor 2 fragment with the inhibitory domain of factor H for the treatment of age-related macular degeneration [116]. However, future and further studies will be essential to evaluate its efficacy for the treatment of kidney diseases, including IRI.

In addition to complement receptors and regulatory proteins that control AP, MASP-2 is an enzyme that has pivotal role in the LP. Dudler et al. developed Narsoplimab, a fully human monoclonal antibody designed to inhibit LP by binding to MASP-2. Narsoplimab is currently in phases II and III of development. Experimental models have demonstrated that the deficiency or inhibition of MASP-2 has beneficial effects in IRI and in the transplantation field, suggesting a potential application in the management of these conditions [117].

### 5.3. Targeting Complement Factors B and D

The selective inhibition of factor B or factor D offers a more targeted approach to block the AP without affecting the innate immunity of the CP and LP. As serine proteases acting upstream of C3, they reduce the activity in the proximal complement amplification loop and the terminal complement pathway. Moreover, their use is expected to lower the risk of infection and other immune-related complications compared to direct C3 inhibition. Currently, oral small-molecule inhibitors targeting complement factors B and D are being developed for use as monotherapy or in combination with a C5 inhibitor [118]. Furthermore, regarding healthcare costs, a cost-effectiveness analysis of a factor B inhibitor demonstrated that it contributes to a higher quality of life for PNH patients at a lower overall cost burden, compared to C5 inhibitors [119].

As shown, the activation of the complement system has been demonstrated in kidneys from deceased-after-brain-death (DBD) donors. Interestingly, C3 deposition was observed on glomerular endothelial cells in kidney biopsies from DBD donors, while no C3 deposition appeared in biopsies from living donors, suggesting that complement deposition is a direct consequence of brain death. In this context, the inhibition of the complement system could be a valuable approach to attenuate renal injury before the transplantation, as shown by Jager et al. They pretreated brain-dead mice with anti-factor B, showing less systemic and local complement system activation, preserved renal function, and attenuated histological injury, suggesting that this might be a promising strategy to minimize complement system-related damage from brain death [120]. Moreover, Amura et al. investigated whether blocking TLRs and the factor B pathway in an animal model of renal IRI could protect against tissue damage. Their findings showed that mice with a targeted deletion of complement factor B alone were protected from injury, while mice with deletion of both factor B and TLR-2 developed more severe damage. This suggests that, although complement and TLR-2 pathways promote pro-inflammatory signals, the selective inhibition of complement factor B may have protective effects against IRI-related damage [121]. Furthermore, Casiraghi et al. demonstrated that factor B-deficient mice underwent reduced IRI, as well as lower T cell-mediated rejection rates and impaired dendritic cell alloreactive functions. These results suggest that factor B inhibitions may have implications for prolonging graft survival, highlighting its potential utility in kidney transplantation [61]. The role of factor D has been explored in a model of gastrointestinal IRI. Stahl et al. showed that factor D-deficient mice exhibited decreased neutrophil accumulation and reduced gastrointestinal injury and suggested that the inhibition of factor D may also represent an effective therapeutic strategy for IRI [122].

Iptacopan is the first drug developed to specifically target complement factor B currently in phase II and III of development for several kidney-involved diseases, including C3 glomerulopathy, IgA nephropaty, lupus nephropathy, and aHUS [123]. Its mechanism of action is based on the inhibition of the enzymatic activity of factor B, which leads to the suppression of both convertases, resulting in an upstream and downstream regulation of the AP [97].

In addition, Danicopan is a first-in-class factor D inhibitor that has been shown to block the initiation of the AP and inhibit up to 80% of downstream events initiated by the classical or lectin pathways through amplification-loop inhibition [124]. A newer oral factor D inhibitor, Vemircopan, is currently under investigation in phase I and II trials for lupus nephritis and IgA nephropathy. While both factor D inhibitors share the same mechanism of action, Vemircopan exhibited greater potency and binding affinity for factor D [125].

## 6. Conclusions and Limitations

This narrative review is intended to highlight that, while many complement-modulating drugs are currently under investigation, only a few of them have been evaluated specifically for their effects on renal IRI, pointing to a significant need for targeted research in this field. One of the potential limitations of starting clinical trials for complement system inhibitors in IRI for kidney transplantation patients is that designing trials involving this specific patient population is complex due to several factors, such as the timing of intervention (pre-, during, or post-IRI), types of donors, and baseline characteristics of recipients, including the underlying cause of kidney disease. Moreover, the interpretation of the data in kidney transplant patients often leads to mixed results. For example, in the context of antibody-mediated rejection, the potential efficacy of complement system-targeting drugs is still not very clear because of the heterogeneous study findings, different drug regimens, and limited sample sizes [126]. The increased susceptibility to infection can be considered another limitation factor for the design of clinical trials in kidney transplant patients, especially given the immunosuppressive therapy required for induction during kidney transplantation. This also raises important ethical concerns about testing these drugs on human subjects. Nonetheless, animal studies can overcome some of the limitations for the use of complement system inhibition strategies in renal transplantation models of IRI, allowing for precise control over experimental conditions, and producing larger sample sizes in order to obtain extensive preclinical data.

The modulation of the complement system has shown promising results in reducing the inflammatory burden, oxidative stress, and tissue damage in many kidney diseases, with a good safety and efficacy profiles. However, studies are still needed to establish their efficacy in renal IRI, and could pave the way for novel therapeutic options in managing kidney injury and improving transplant success. By temporarily “switching off” complement system activation during the acute injury phases, these drugs may enhance renal protection, especially since the kidney itself is capable of synthetizing complement proteins [96]. Moreover, the penetrance of these molecules in the renal interstitium makes them attractive therapeutic approaches in the field of transplantation.

## Figures and Tables

**Figure 1 antioxidants-14-00066-f001:**
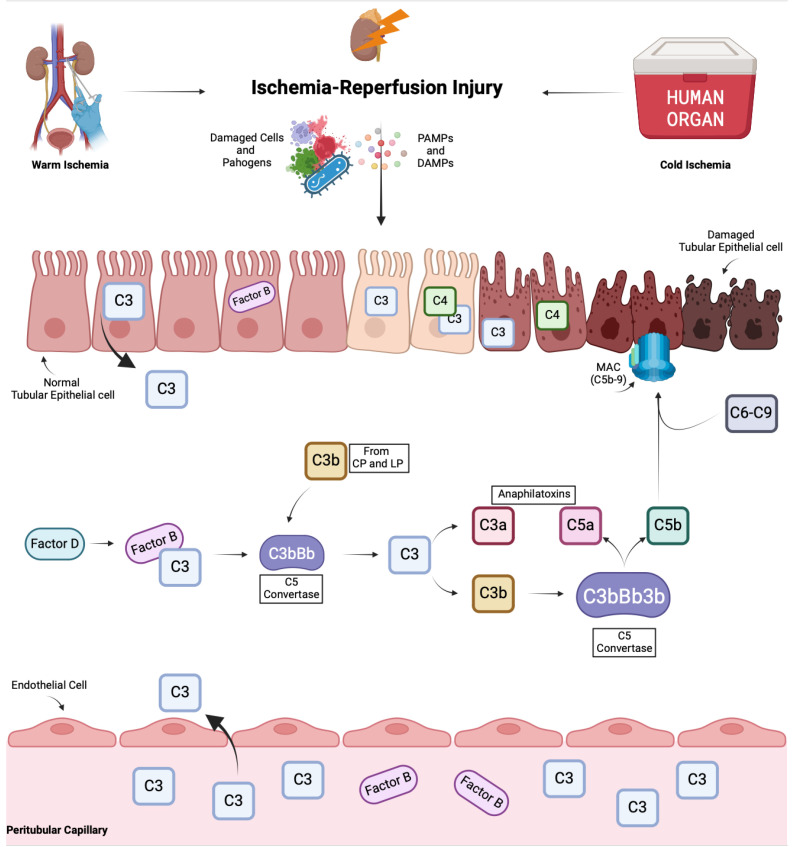
The kidneys are particularly susceptible to ischemia–reperfusion injury (IRI), especially in the context of transplantation, where exposure to warm and cold ischemic periods can trigger IRI. The exposition to ischemic and reperfusion phases can lead to the release of damage-associated molecular patterns (DAMPs) (and potentially pathogen-associated molecular patterns, PAMPs), which may activate the complement system. Particularly, after IRI, renal epithelial cells (RECs) undergo functional and structural changes, which leads to the amplification of the complement system and production of complement components. The alternative pathway is especially involved in the initiation of the complement cascade, leading to the formation of the membrane attack complex (MAC, C5b-9) and the release of anaphylatoxins, both contributing to cell and tissue damage. Created in BioRender.

**Figure 2 antioxidants-14-00066-f002:**
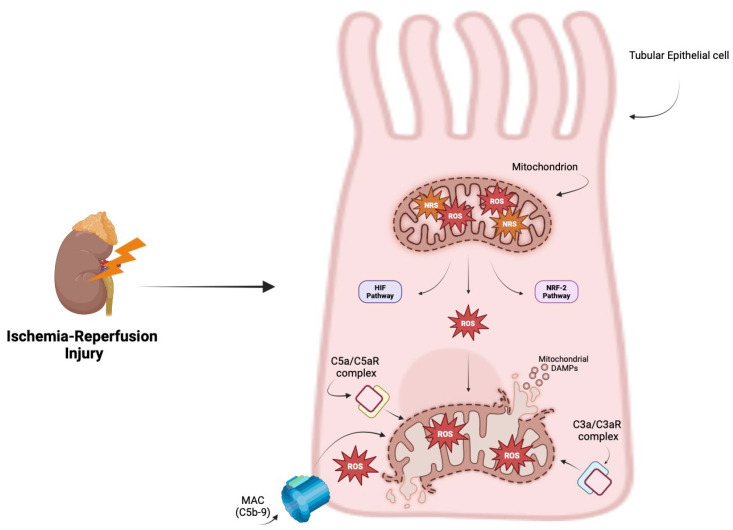
Ischemia–reperfusion injury triggers the production of reactive oxygen species (ROS) and reactive nitrogen species (RNS), resulting in rapid tissue damage. Mitochondria are considered the primary source of ROS; they activate pivotal pathways involved in OxS responses, including the hypoxia-inducible factor (HIF) and nuclear factor erythroid2-related factor2 (NRF2) pathways, as adaptive responses to mitigate the OxS-related damage. Moreover, OxS has been shown to activate the complement system, leading to the formation of membrane attack complex (MAC) on cells, which induces additional OxS burden. Anaphylatoxins, especially the C3a/C3aR and C5a/C5aR pathways, have also been implicated in the impairment of mitochondrial functions, sensitizing cells to ROS-induced damage. Created in BioRender.

**Figure 3 antioxidants-14-00066-f003:**
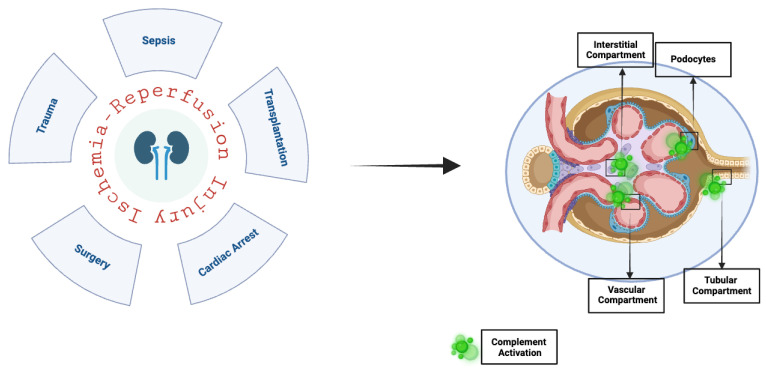
In the kidneys, ischemia–reperfusion injury can be triggered by several factors, such as surgical procedures, sepsis, trauma, cardiac arrest, and transplantation, leading to the activation of the complement system, which plays a key role in mediating the damage. The complement activation occurs across all kidney compartments, including the vascular, glomerular, and interstitial compartments, contributing to widespread tissue injury and dysfunction. Created in BioRender.

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
