# Peer review of "Exploring Potential Complement Modulation Strategies for Ischemia–Reperfusion Injury in Kidney Transplantation"

_antioxidants, 2025, doi:10.3390/antiox14010066_

Round 1
Reviewer 1 Report
The present study investigates the potential of complement modulation strategies for ischemia/reperfusion injury in kidney transplantation. I have the following questions below.
Could you explain why the interactions between the complement system and other pathways related to ischemia-reperfusion injury, like coagulation or TLR signaling, were not explored?
What strategies would you recommend to reduce the risks associated with complement-targeting drugs in immunocompromised transplant recipients, including increased infection risk or unintended side effects?
Are there specific biomarkers or clinical indicators you would suggest for identifying patients who might gain the most from therapies that modulate the complement system?
Could you elaborate on how complement-targeting therapies might be combined with other treatments, such as antioxidants or anti-inflammatory drugs, to address the multifactorial nature of IRI?
Why does the publication not include any specific ongoing or planned clinical trials for complement inhibitors in IRI for kidney transplantation? What are the challenges to starting such trials?
The study discusses potential combinations of C3 and C5 inhibitors. Are there any preclinical or clinical studies supporting this technique, and how do you plan to overcome the obstacles of dual inhibition?
Could you explain how other targets, such as Factor B or Factor D, may provide advantages to C3 and C5 inhibition in terms of efficacy, safety, and cost?
Given the overreliance on animal studies, how do you explain the lack of human data in your conclusions about the efficacy and safety of complement-targeting therapies?
Given the differences in complement activation profiles and IRI risks, how would you make therapeutic recommendations for living versus deceased donor kidney transplants?
Author Response
We thank the reviewer for the suggestions, we have incorporated them into the changes we've made.
- Comment 1: Could you explain why the interactions between the complement system and other pathways related to ischemia-reperfusion injury, like coagulation or TLR signaling, were not explored?
- Response: Thank you for the time dedicated to revise our manuscript and for pointing this out. We’ve intended to conduct a narrative review that focuses on providing an overview of pharmacological agents that act as modulators of the complement system and that are currently being tested or could be tested to prevent renal damage caused by ischemia/reperfusion after transplantation. Our review includes a description of the complement system, its relationship with oxidative stress, and its involvement in renal injury during IRI. While we chose not to delve deeply into the complexity of interactions between the several pathways related to ischemia-reperfusion injury, we acknowledge their significant role in mediating the Therefore, we have integrated, at least in part, references to these pathways in the text. (Lines 128-153)
- Comment 2: What strategies would you recommend to reduce the risks associated with complement-targeting drugs in immunocompromised transplant recipients, including increased infection risk or unintended side effects?
Response: We sincerely thank the reviewer for this suggestion. There is evidence that the widespread use of vaccines prior to the initiation of complement inhibitor therapy, along with the prophylactic use of antibiotics and other antimicrobial agents, as well as the temporary and selective inhibitions of the complement system, can significantly reduce the risks of infections and unintended side effects. We’ve revised the manuscript to incorporate these considerations. (Lines 168-190)
- Comment 3: Are there specific biomarkers or clinical indicators you would suggest for identifying patients who might gain the most from therapies that modulate the complement system?
Response: We thank the reviewer for the input. Given the critical role of the complement system in renal IRI, monitoring complement components and fractions, as well as inflammatory and endothelial dysfunction markers, could guide the therapeutic approaches in order to tailor complement-targeting therapies to individual patients and optimize clinical outcomes. We’ve revised the manuscript to reflect these insights. (Lines 390-412)
- Comment 4: Could you elaborate on how complement-targeting therapies might be combined with other treatments, such as antioxidants or anti-inflammatory drugs, to address the multifactorial nature of IRI?
Response: We thank the reviewer for the suggestions; we’ve integrated them into the manuscript. (Lines 285-307)
- Comment 5: Why does the publication not include any specific ongoing or planned clinical trials for complement inhibitors in IRI for kidney transplantation? What are the challenges to starting such trials?
Response: Thank you for pointing this out. Designing trials involving this specific patient population is complex due to several factors, including the timing of intervention, the type of kidney donors, and the baseline characteristics of the recipients. We’ve discussed potential limitations in the text. (Lines 685-700)
- Comment 6: The study discusses potential combinations of C3 and C5 inhibitors. Are there any preclinical or clinical studies supporting this technique, and how do you plan to overcome the obstacles of dual inhibition?
Response: We thank the reviewer for the comment. Unfortunately, there are no preclinical or clinical studies about renal IRI that we are aware of, however the effects of the combination of C3 and C5 inhibition have been explored in paroxysmal nocturnal hemoglobinuria patients (https://doi.org/10.1016/S2352-3026(21)00028-4, https://doi.org/10.1182/blood.2021011388). It seems that one of the major deterrent to the combined use of a licensed C5 inhibitor and a licensed C3 inhibitor is financial as stated by Notaro et al (https://doi.org/10.1056/NEJMra2201664) . We think that close monitoring of complement activation markers, prophylactic vaccinations, regular infection screenings will be mandatory to reduce the side effects of the dual complement inhibition. We’ve integrated the suggestions in the manuscript. (Lines 529-544) - Comment 7: Could you explain how other targets, such as Factor B or Factor D, may provide advantages to C3 and C5 inhibition in terms of efficacy, safety, and cost?
Response: Thank you for the input. In terms of efficacy, the selective inhibition of Factor B or Factor D offers a more targeted approach to block the Alternative Pathway without affecting the innate immunity of the Classical and Lectin Pathways, therefore providing an upstream control over complement activation and a reduction of the proximal complement amplification loop. From a safety perspective, such a selective binding and inhibition profile may mitigate safety concerns associated with inhibition of all downstream complement activity. Moreover, their use is expected to lower the risk of infection and other immune-related complications compared to direct C3 inhibition. Regarding costs, Factor B and Factor D inhibitors are usually small molecules, in pill form, and they can be distributed by retail. Contrarily, monoclonal antibodies are usually administered directly to the patient in a hospital, increasing the associated healthcare costs (doi: https://doi.org/10.1016/j.medidd.2020.100075). A cost-effectiveness analysis of a factor B inhibitor (Iptacopan) demonstrated that iptacopan contributes to a higher quality of life for PNH patients at a lower overall cost burden, as compared to C5 inhibitors (https://doi.org/10.1182/blood.2024025176). We’ve incorporated your suggestions into Lines 635-643. - Comment 8: Given the overreliance on animal studies, how do you explain the lack of human data in your conclusions about the efficacy and safety of complement-targeting therapies?
Response: Thank you for your comment. We’ve revised the conclusion and limitations paragraph accordingly. (Lines 685-700) - Comment 9: Given the differences in complement activation profiles and IRI risks, how would you make therapeutic recommendations for living versus deceased donor kidney transplants?Response: We are deeply grateful for the time you dedicated to reviewing our manuscript and your suggestions that have significantly improved the quality of the manuscript. Based on the current knowledge and the absence of established guidelines for complement-targeting therapies in this setting of patients, it is reasonable to hypothesize that therapeutic approaches may differ between living and deceased donors due to the different risks of IRI. Deceased donors often experience prolonged cold ischemia times leading to more severe IRI, in this context the inhibition of the complement system before the transplantation could help to attenuate renal injury (lines 646-654). Conversely, living kidney donors typically experience less severe IRI, allowing for a more tailored complement modulating therapies, particularly for high-risk cases such as donors with preexisting immune sensitization or high levels of complement activation markers.
Reviewer 2 Report
Dario Troise et al. have conducted an extensive and comprehensive narrative review in which they present, as the most important aspect, an overview of pharmacological agents that act as modulators of the complement system and that are currently being tested or could be tested to prevent renal damage caused by ischemia/reperfusion after transplantation.
From my point of view, I think that this is a very complete review article that reflects the current situation of this type of therapies (many of them are very novel and are currently being tested).
The work consists of some introductory sections that describe in detail the complement system, its relationship with oxidative stress and its involvement in renal damage due to ischemia/reperfusion. The bulk of the article, on the other hand, deals with the different pharmacological groups (classified by their mechanism of action on the complement system) under evaluation (describing clinical studies and analyses and summarizing their results).
The paper concludes that only a small part of these complement modulating agents have been evaluated to prevent renal damage due to ischemia/reperfusion, and that it would be appropriate to test them in this type of injury because they could provide satisfactory results.
The article also includes illustrative figures of high scientific quality elaborated by the authors.
I have not found any aspect that needs to be corrected.
Author Response
- Comment 1: Dario Troise et al. have conducted an extensive and comprehensive narrative review in which they present, as the most important aspect, an overview of pharmacological agents that act as modulators of the complement system and that are currently being tested or could be tested to prevent renal damage caused by ischemia/reperfusion after transplantation. From my point of view, I think that this is a very complete review article that reflects the current situation of this type of therapies (many of them are very novel and are currently being tested). The work consists of some introductory sections that describe in detail the complement system, its relationship with oxidative stress and its involvement in renal damage due to ischemia/reperfusion. The bulk of the article, on the other hand, deals with the different pharmacological groups (classified by their mechanism of action on the complement system) under evaluation (describing clinical studies and analyses and summarizing their results). The paper concludes that only a small part of these complement modulating agents have been evaluated to prevent renal damage due to ischemia/reperfusion, and that it would be appropriate to test them in this type of injury because they could provide satisfactory results. The article also includes illustrative figures of high scientific quality elaborated by the authors. I have not found any aspect that needs to be corrected.
Response: We sincerely thank the reviewer for the flattering comments on our manuscript and are pleased that it has met the reviewer’s expectations for publication in the Antioxidants Journal
Reviewer 3 Report
This is an up-to-date review on complement system role in IRI, going from the bench molecular background to the bedside implications for therapy of acute injury or prevention of its consequences.
The detailed analysis of recent and ongoing clinical trials gives a perspective on future potential utilities of complement system blockade on various levels, concerning complement elements, their receptors, or inhibitory factors.
The subject of complement activation/blockade in hemolytic uremic syndrome, being an excellent example of acute kidney injury, should be extended beyond a single sentence. C5 blockade in patients with HUS, as well as subsequent preventive eculizumab/rawulizumab administration before planned kidney transplantation, with continuation of the therapy after RTx, deserves attention also because of essential questions - when to start C5 blockade and when to finish it.
Author Response
We thank the Reviewer for the suggestions, we've incorporated them into the changes we made.
- Comment 1: The subject of complement activation/blockade in hemolytic uremic syndrome, being an excellent example of acute kidney injury, should be extended beyond a single sentence. C5 blockade in patients with HUS, as well as subsequent preventive eculizumab/rawulizumab administration before planned kidney transplantation, with continuation of the therapy after RTx, deserves attention also because of essential questions - when to start C5 blockade and when to finish it.
Response: We are deeply grateful for the time dedicated to reviewing our manuscript. We’ve intended to conduct a narrative review that focuses on providing an overview of pharmacological agents that act as modulators of the complement system and that are currently being tested or could be tested to prevent renal damage caused by ischemia/reperfusion after transplantation. Our review includes a description of the complement system, its relationship with oxidative stress, and its involvement in renal injury during IRI. While we chose not to delve deeply into the complexity of the activation of the complement system in other diseases, such as hemolytic uremic syndrome, we acknowledge its significant role in mediating the damage. Therefore, we have incorporated the reviewer’s suggestions into the manuscript, albeit briefly. (Lines 502-511)
Round 2
Reviewer 1 Report
The paper can be accepted in its present form.
The paper can be accepted in its present form.